# Structural basis for the interaction of human herpesvirus 6B tetrameric glycoprotein complex with the cellular receptor, human CD134

Mitsuhiro Nishimura[1], Bernadette Dian Novita[1,2], Takayuki Kato[3], Lidya Handayani Tjan[1], Bochao Wang[1], Aika Wakata[1], Anna Lystia Poetranto[1], Akiko Kawabata[1], Huamin Tang[1], Taiki Aoshi[4], Yasuko Mori[1] *

1 Division of Clinical Virology, Center for Infectious Diseases, Kobe University Graduate School of Medicine, Kobe, Hyogo, Japan, 2 Department of Pharmacology and Therapy, Faculty of Medicine, Widya Mandala Catholic University, Surabaya, Indonesia, 3 Protonic NanoMachine Group, Graduate School of Frontier Biosciences, Osaka University, Suita, Osaka, Japan, 4 Vaccine Dynamics Project, BIKEN Innovative Vaccine Research Alliance Laboratories, Research Institute for Microbial Diseases (RIMD), Osaka University, Suita, Osaka, Japan

* ymori@med.kobe-u.ac.jp

**Data Availability Statement:** Crystal structure of the Fab fragment of murine monoclonal antibody KH-1 against Human herpesvirus 6B DOI: https://

## Abstract

A unique glycoprotein is expressed on the virus envelope of human herpesvirus 6B (HHV-6B): the complex gH/gL/gQ1/gQ2 (hereafter referred to as the HHV-6B tetramer). This tetramer recognizes a host receptor expressed on activated T cells: human CD134 (hCD134). This interaction is essential for HHV-6B entry into the susceptible cells and is a determinant for HHV-6B cell tropism. The structural mechanisms underlying this unique interaction were unknown. Herein we solved the interactions between the HHV-6B tetramer and the receptor by using their neutralizing antibodies in molecular and structural analyses. A surface plasmon resonance analysis revealed fast dissociation/association between the tetramer and hCD134, although the affinity was high ($K_D$ = 18 nM) and comparable to those for the neutralizing antibodies (anti-gQ1: 17 nM, anti-gH: 2.7 nM). A competition assay demonstrated that the anti-gQ1 antibody competed with hCD134 in the HHV-6B tetramer binding whereas the anti-gH antibody did not, indicating the direct interaction of gQ1 and hCD134. A single-particle analysis by negative-staining electron microscopy revealed the tetramer's elongated shape with a gH/gL part and extra density corresponding to gQ1/gQ2. The anti-gQ1 antibody bound to the tip of the extra density, and anti-gH antibody bound to the putative gH/gL part. These results highlight the interaction of gQ1/gQ2 in the HHV-6B tetramer with hCD134, and they demonstrate common features among viral ligands of the betaherpesvirus subfamily from a macroscopic viewpoint.

doi.org/10.2210/pdb6LKT/pdb Crystal structure of the Fab fragment of murine monoclonal antibody OHV-3 against Human herpesvirus 6B DOI: https://doi.org/10.2210/pdb6LTG/pdb

**Funding:** Y.M. was supported by Acceleration Transformative research for Medical innovation (ACT-MS) from Japan Agency for Medical Research and Development (AMED) under Grant Number JP17im0210601, https://www.amed.go.jp/. M.N. was supported by JSPS KAKENHI Grant-in-Aid for Scientific Research (C) under Grant Number JP19K06512, https://www.jsps.go.jp/; Hyogo Science and Technology Association, http://hyogosta.jp/; and Takeda Science Foundation, https://www.takeda-sci.or.jp/. The funders had no role in study design, data collection and analysis, decision to publish, or preparation of the manuscript.

**Competing interests:** The authors declare no conflicts.

## Author summary

Primary infection of human herpesvirus 6B (HHV-6B) with fever and roseola occurs for almost all children, and HHV-6B remains in the host as a latent infection. The reactivation of HHV-6B (especially in patients after hematopoietic stem-cell transplantation) occasionally causes severe encephalitis, which is a public health concern worldwide. HHV-6B's unique gH/gL/gQ1/gQ2 complex (a tetramer expressed on the viral envelope) recognizes the cellular receptor, i.e., human CD134 (hCD134), which is essential for the entry of the virus into the host's cells. The interaction between HHV-6B tetramer and hCD134 is therefore one of the determinants of HHV-6B-specific cell tropism. This article sheds light on the molecular and structural interactions between HHV-6B tetramer and its host receptor along with their neutralizing antibodies, and their affinities, competition, and binding modes based on the HHV-6B tetramer structure are described. Our findings provide molecular and structural bases for a comprehensive understanding of these interactions and relationships.

## Introduction

Human herpesvirus 6B (HHV-6B) is the major causative pathogen of the exanthema subitum contracted during infancy, with high fever followed by a skin rash [1]. Although the prognosis of HHV-6B infection is benign in most cases, the infection sometimes progresses to encephalitis, leaving neurological sequelae [2]. Like other herpesviruses, the primary infected HHV-6B virus establishes latency and persists in the body as long as the host lives. The latent HHV-6B virus reactivates in response to stimulation such as immune-compromised conditions, drugs, and fatigue. In immunocompromised patients — especially recipients of hematopoietic stem-cell transplantation — the latent HHV-6B reactivates, proliferates, and finally progresses to severe encephalitis at a high rate [2,3,4,5].

A human host's immune system is able to prevent HHV-6B infection. Infants are believed to be protected from HHV-6B infection by maternal antibodies, since the infection occurs at the ages of 6 months to 2 years, which coincides with the ages at which maternal immunity declines [6,7]. Considering the fact that almost all adults worldwide are seropositive for HHV-6B, this infection is an inevitable event in human life; its potential risk is common to all human beings. It is therefore of importance to understand how HHV-6B is controlled by a host's immunity.

HHV-6B belongs to the betaherpesvirus subfamily in the *Herpesviridae* family, and it was distinguished from human herpesvirus 6A (HHV-6A) based on the two viruses' significant differences in pathogenicity, cell tropism, and other characteristics [8,9,10]. HHV-6B has shown similarity with other members of the betaherpesvirus subfamily, i.e., human herpesvirus 7 (HHV-7) and human cytomegalovirus (HCMV). Core genes of a herpesvirus are shared with other herpesviruses including herpes simplex virus type I and II (HSV-1 and HSV-2), varicella zoster virus (VZV), Epstein-Barr virus (EBV), and Kaposi sarcoma human herpesvirus (KSHV).

One of the critical differences in these herpesviruses is the target cells that they enter and infect, which leads to their own cell tropisms. Herpesviruses use a fundamental set of envelope glycoproteins (gB and gH/gL) on the virion surface to initiate the infection to target cells [11,12]. The homotrimeric fusogenic protein gB mediates the viral-host membrane fusion that is required for putting the nucleocapsid into the host cytosol. In contrast, gH/gL is a heterodimeric complex tethered on the virion membrane via the C-terminal transmembrane domain

of gH. The gH/gL complex required for the activation of gB to exert the fusion function, but the details of the cooperation between gB and gH/gL have been unclear.

Each herpesvirus has additional envelope glycoproteins that engage with respective host receptors, thereby differentiating the cell tropisms. It was revealed that the gH/gL complex of HHV-6B and HHV-6A are associated with the unique glycoproteins gQ1 and gQ2 encoded in the open reading frame (ORF) U100, resulting in a tetrameric complex gH/gL/gQ1/gQ2, while a trimeric complex gH/gL/gO is also known including another glycoprotein gO encoded in the ORF U47 [13,14,15,16,17]. The gH/gL/gQ1/gQ2 complex (hereafter referred to as the 'tetramer') is especially important because gQ1 and the associated gQ2 play critical roles in the interaction with the host receptors [14,18,19,20].

HHV-6A and HHV-6B recognize their specific receptors by differentiating gQ1 and gQ2; the HHV-6A tetramer binds to CD46, which is widely expressed on human cells [21], whereas the HHV-6B tetramer recognizes human CD134 (hCD134, also known as OX40) which is specifically expressed on activated T cells [19]. The responsible residues in both HHV-6B gQ1 and hCD134 were identified in previous studies, but the interaction is not simple as it depends on the conformation of gQ1 in cooperation with gQ2, and in addition with gH/gL [18,20].

The results of structural analyses of the gH/gL heterodimer and gH/gL-based glycoprotein complexes with additional viral components have been published for several human herpesviruses: crystal structures at atomic resolution have been reported for HSV-2 [22], VZV [23], EBV (gH/gL and gH/gL/gp42) [24,25], and HCMV (gH/gL/UL128/UL130/UL131; hereafter referred to as the 'pentamer')[26]. The gH/gL structure shares a common arrangement in which the N-terminal region of gH is combined with the gL, forming one tip of its elongated shape, and the subsequent part of gH extends toward the opposite tip. The C-terminal region of gH contains a single-pass transmembrane domain, and thus the gH/gL heterodimer is expected to place the gH C-terminal region at a membrane-proximal side, with the gL located at a membrane-distal side. Both the EBV gH/gL/gp42 and the HCMV pentamer extend the gH/gLs by combining with associated factors that are essential for binding to specific receptors. However, the relative arrangement between gH/gL and the associated factors are strikingly different. The tetramer structures of HHV-6B and HHV-6A are largely unknown because of the lack of information about the unique glycoproteins gQ1 and gQ2.

Antibodies against the surface antigens of viruses play major roles in protection from the viral infection. The immunization of mice with inactivated HHV-6B elicited neutralizing antibodies, and interestingly, most of the monoclonal antibodies (Mabs) with neutralizing potency target the tetramer [27]. An HHV-6B-specific neutralizing Mab for gQ1 named KH-1 was analyzed in detail [27,28]. The neutralizing potency against HHV-6B infection has also been shown for a Mab for gH, OHV-3 [28,29]. One possible scenario regarding the mechanism of Mabs for the neutralization of gQ1 is as follows: the access of the gQ1 in the tetramer to the host receptor hCD134 is blocked and the gQ1 itself cannot bind to hCD134. However, the underlying mechanism had not been established. In the present study, the interactions between the HHV-6B tetramer and host receptor with neutralizing antibodies were solved from the molecular and structural perspectives.

## Results

### Affinity of the tetramer and hCD134 or neutralizing antibodies

Despite the critical importance of the molecular interaction between the HHV-6B gH/gL/gQ1/gQ2 complex (i.e., the 'tetramer') and the host receptor hCD134 for HHV-6B infection, the affinity of the tetramer and receptor has not been evaluated. We therefore constructed the soluble HHV-6B tetramer for analysis. A mammalian expression system was used for the

tetramer's construction because the complicated hetero-tetrameric complex is built up during its travel through the host's organella while undergoing glycosylation [13,15]. Using the mammalian system, we established a stable expression system of the soluble tetramer by excluding the transmembrane domain in the gH. The tetramer was purified to homogeneity (S1 Fig) and used herein for the following analyses.

We performed a surface plasmon resonance analysis to quantitatively evaluate the affinity between the soluble tetramer and hCD134 or neutralizing antibodies for HHV-6B infection. As the analyte, each of anti-gQ1 Mab KH-1, anti-gH Mab OHV-3, and the recombinant hCD134 with C-terminal human FcHis-tag replacing the transmembrane domain (hCD134-hFcHis) was separately immobilized on a sensor chip and the soluble tetramer was loaded. The response curves for the anti-gQ1 Mab KH-1/tetramer interaction and the anti-gH Mab OHV-3/tetramer interaction were collected by a single-cycle injection method (Fig 1A and 1B). The calculated kinetic parameters are shown in Table 1.

The affinity of the anti-gQ1 Mab KH-1 and that of anti-gH Mab OHV-3 for the tetramer were substantially high, as the dissociation constant $K_D$ values were 17 nM and 2.7 nM, respectively. The curves' shapes and the kinetics parameters indicated slow association and dissociation rates between the tetramer and each Mab (Fig 1A and 1B and Table 1). In contrast, the interaction with hCD134-hFcHis showed a fast dissociation pattern, and a single-cycle kinetics analysis was not applicable. We thus assessed the affinity by using a multiple-cycle injection method. The shape of the curve indicated fast association and dissociation rates (Fig 1C and Table 1) in contrast to those for the Mabs, although the analyzed affinity was high, i.e., $K_D$ = 14 nM, which is comparable to the affinity for the Mabs. Because the fast association and dissociation resulted in slightly poor fitting quality of the kinetics analysis (Fig 1C and Table 1), we used a steady-state affinity model to determine the dissociation constant; the affinity was $K_D$ = 18 nM (Fig 1D), as a more reliable value.

## Competition between hCD134 and each neutralizing Mab

A simple and intuitive mechanism of neutralizing antibodies is to prevent the engagement between the virus ligand and the host receptor. Because both hCD134 and Mab KH-1 recognize the gQ1 of the tetramer [20,27] and since they have similar affinity as revealed above (Table 1), competition between them is expected. To test this expectation, we conducted a competition experiment based on an enzyme-linked immunosorbent assay (ELISA). The binding of the hCD134-hFcHis to the tetramer coated on the plate was subjected to competition with the anti-gQ1 Mab KH-1 and with the anti-gH Mab OHV-3. The amount of hCD134-hFcHis bound on the tetramer was decreased as the concentration of anti-gQ1 Mab KH-1 increased (Fig 2A), indicating that anti-gQ1 Mab KH-1 competed with hCD134-hFcHis for binding to the tetramer. The signal was decreased in a concentration-dependent manner from the unaffected level detected without Mab (Fig 2A, filled arrowhead) to the baseline level detected without hCD134-hFcHis at the Mab concentration of 2.4 µg/ml (Fig 2A, open arrowhead). In contrast, the anti-gH Mab OHV-3 did not inhibit the binding of hCD134-hFcHis to the tetramer. The signal stayed at the unaffected level even at the highest concentration, 2.4 µg/ml (Fig 2B). The anti-gH Mab OHV-3 bound to the tetramer in the presence of hCD134-hFcHis (S2A Fig), indicating that the hCD134-hFcHis and the anti-gH Mab OHV-3 are able to bind to the tetramer independently. In previous study, we have demonstrated that the anti-gH Mab OHV-3 had the neutralizing activity only in a high concentration range ($IC_{50}$ 7.7 µg/ml) [28], thus the same ELISA experiment as Fig 2B was done at a higher concentration range up to 80 µg/ml. The anti-gH Mab OHV-3 could not inhibit the hCD34-hFcHis binding even at the highest concentration 80 µg/ml (S2B Fig).

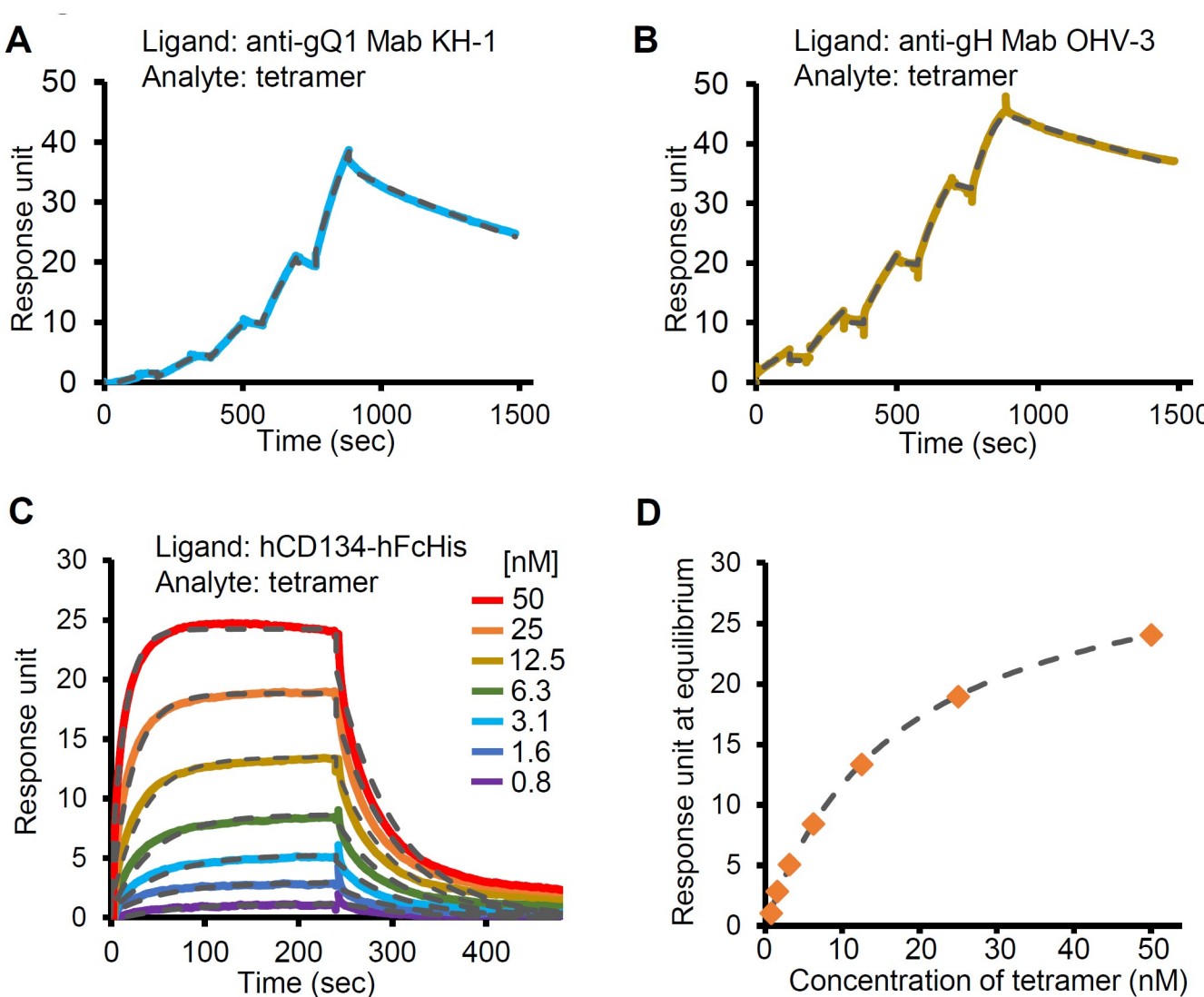

**Fig 1. Surface plasmon resonance analysis between the tetramer and the host receptor or neutralizing Mabs for HHV-6B.** The sensorgrams of the loaded tetramer to the anti-gQ1 Mab KH-1 (A) or anti-gH Mab OHV-3 (B) immobilized on the CM5 chip are shown. The concentration of the loaded tetramer was serially increased as 5, 10, 20, 40, and 80 nM. The fitting curves analyzed by the single-cycle kinetics method are depicted as *dashed lines*. (C) The multi-cycle sensorgram for the binding of the loaded tetramer to the immobilized hCD134-hFcHis. The concentration of the loaded tetramer was increased as indicated. The fitting curves analyzed by the multi-cycle kinetics method based on 1:1 binding are shown as *dashed lines*. (D) The response units at the equilibrium are plotted for the tetramer concentration. *Dashed line*: The fitting curve obtained by the steady-state affinity analysis.

**Table 1. The affinity of the tetramer for the antibodies or the receptor.**

| Ligand | $k_a$, M$^{-1}$, s$^{-1}$ (×10$^5$) [†] | $k_d$, s$^{-1}$ (×10$^{-4}$) [†] | $K_D$ (nM) [†] | $\chi^2$ (RU$^2$)* |
|---|---|---|---|---|
| Anti-gQ1 Mab KH-1 | 0.39 ± 0.002 | 6.7 ± 0.02 | 17 | 0.0891 |
| Anti-gH Mab OHV-3 | 1.3 ± 0.001 | 3.5 ± 0.01 | 2.7 | 0.1599 |
| hCD134-hFcHis | | | | |
| Multi-cycle kinetics | 11 ± 0.746 | 158 ± 0.62 | 14 | 1.0429 |
| Steady-state affinity | – | – | 18 ± 1.1 | 0.0526 |

*The $\chi^2$-test value for each fitting curve.

†Standard deviations (±SD) of the fitting analysis.

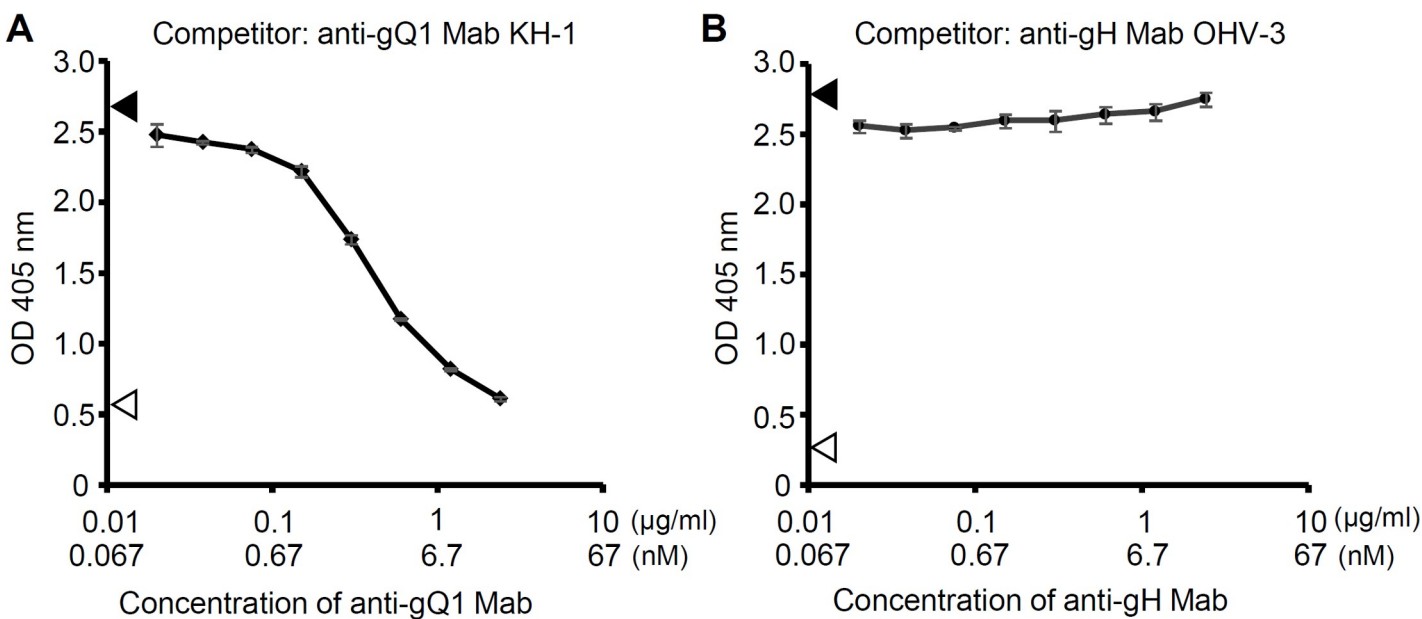

**Fig 2. Competition between hCD134 and anti-gQ1 or anti-gH Mab for the tetramer binding.** The binding of hCD134-hFcHis to the tetramer coated on the plate was detected by an ELISA, and the competition between hCD134-hFcHis and anti-gQ1 Mab KH-1 (A) or anti-gH Mab OHV-3 (B) was analyzed. The concentration of the hCD134-hFcHis was 0.1 μg/ml, and the Mab concentration was varied at 0.02, 0.038, 0.075, 0.15, 0.3, 0.6, 1.2 and 2.4 μg/ml as the final concentration. The plotted points are the averages of three wells in the same condition. *Bars*: SD of the wells. The *filled arrowheads* and *open arrowheads* indicate the values without Mab and those without hCD134-hFcHis in the presence of 2.4 μg/ml Mab, respectively; one of the duplicate results is shown for each.

### Determination of the antigen binding sites of the tetramer with each neutralizing Mab

We determined the sequences of the complementarity-determining regions (CDRs) of anti-gQ1 Mab KH-1 and anti-gH Mab OHV-3 in our previous study [28], but their spatial arrangements remained to be elucidated. Herein, the Fab domains of the anti-gQ1 Mab KH-1 and those of the anti-gH Mab OHV-3 were prepared by papain digestion, and the crystal structures were determined. The crystallographic statistics are presented in Table 2. The structure and the arrangement of the CDRs of each Fab are depicted in Fig 3. The Fab domain of anti-gQ1 Mab KH-1 presented a positively charged area at the cleft between the heavy chain and light chain, including a total of eight positively charged residues (Fig 3B and 3C, blue). The other notable feature was the seven tyrosine residues located along the cleft (Fig 3C, brown).

In contrast, the Fab domain of anti-gH Mab OHV-3 (Fig 3D) has a negatively charged electrostatic potential at the cleft (Fig 3E) due to the contribution of two glutamic acid residues (Fig 3F, red). The residues 100–105 at the CDR-H3 were not determined due to the poor electron density (Fig 3D and 3E and Fig 3F, underline). The remaining surface is hydrophobic, with two tryptophan residues and one valine residue located near the negatively charged area as well. One tyrosine residue was also close to the center of the antigen binding site.

### Molecular and structural shapes of the tetramer visualized by negative-staining electron microscopy

The HHV-6B tetramer consists of the gH/gL heterodimer (which is common among herpesviruses) and the additional components gQ1/gQ2 (which is unique to HHV-6B and HHV-6A). Although gH/gL heterodimer is likely to have a structure that is comparable to those of the known herpesviruses, there is no available information about gQ1 and gQ2 or the overall

**Table 2. Results of the X-ray crystallographic analysis of the Fab domains of the neutralizing Mabs.**

| Parameter | Anti-gQ1 Fab KH-1 | Anti-gH Fab OHV-3 |
|---|---|---|
| Data collection: | | |
| Wavelength (Å) | 1.000000 | 1.000000 |
| Space group | P1 21 1 | P21 21 21 |
| Unit cell: | | |
| a, b, c (Å) | 39.4, 120.0, 42.6 | 81.4, 95.2, 113.78 |
| α, β, γ (°) | 90.0, 97.0, 90.0 | 90.0, 90.0, 90.0 |
| Resolution (Å) | 39.1–1.80 (1.86–1.80) | 38.3–1.63 (1.69–1.63) |
| Unique reflections | 36,273 (2,610) | 110,577 (10,908) |
| R-merge | 0.086 (0.482) | 0.064 (0.75) |
| R-meas | 0.093 (0.674) | 0.069 (0.81) |
| Mean I/sigma (I) | 12.81 (2.78) | 13.4 (1.85) |
| Completeness (%) | 99.9 (98.8) | 99.9 (100.0) |
| Multiplicity | 7.45 (6.77) | 3.56 (3.65) |
| $CC_{1/2}$ | 99.9 (87.1) | 99.9 (81.4) |
| Refinement: | | |
| R-work | 0.165 (0.223) | 0.182 (0.274) |
| R-free | 0.196 (0.250) | 0.204 (0.301) |
| Nos. of atoms: | | |
| Macromolecules | 3,240 | 6586 |
| Water | 487 | 1332 |
| Average B-factor ($Å^2$): | 22.1 | 23.9 |
| Macromolecules | 20.6 | 21.8 |
| Solvent | 32 | 33.9 |
| RMS deviations: | | |
| Bonds (Å) | 0.007 | 0.007 |
| Angles(°) | 1.10 | 1.26 |
| Ramachandran (%): | | |
| Favored | 99.0 | 96.5 |
| Outliers | 0.0 | 0.0 |

*The data for the highest-resolution shell are presented in parentheses.

arrangement as the tetrameric complex. To unveil the overall shape of the tetramer, we subjected the purified soluble tetramer to an analysis using electron microscopy by a negative-staining method. The electron microscopy visualized the dispersed particles of the tetramer (Fig 4A). To further analyze the molecular shape of the tetramer, we performed a single particle analysis. A two-dimensional (2D) classification analysis of the tetramer images revealed an elongated structure of the tetramer with a large bend at the mid-position (Fig 4B).

In a representative view of the density, the two parts of the bent shape were approx. 120 Å and 100 Å in length, and the angle at the bend was approx. 90° (Fig 4C). After the selection of 65,663 good particles from 2D class average, we reconstructed the three-dimensional (3D) density from the particles at a 17.4 Å resolution, confirming the elongated-curved shape of the tetramer (Fig 4D). The 3D structure consists of a relatively flattened part that we named the 'Palm' and a cylindrical part we named the 'Stalk' (Fig 4C and 4D). The Stalk also showed a shallow bend near the end area (Fig 4C, arrow), which may correspond to that of the "Heel-Sole-Toe" arrangement observed in the gH structures of HSV-2, VZV, and HCMV [12,22,23,26].

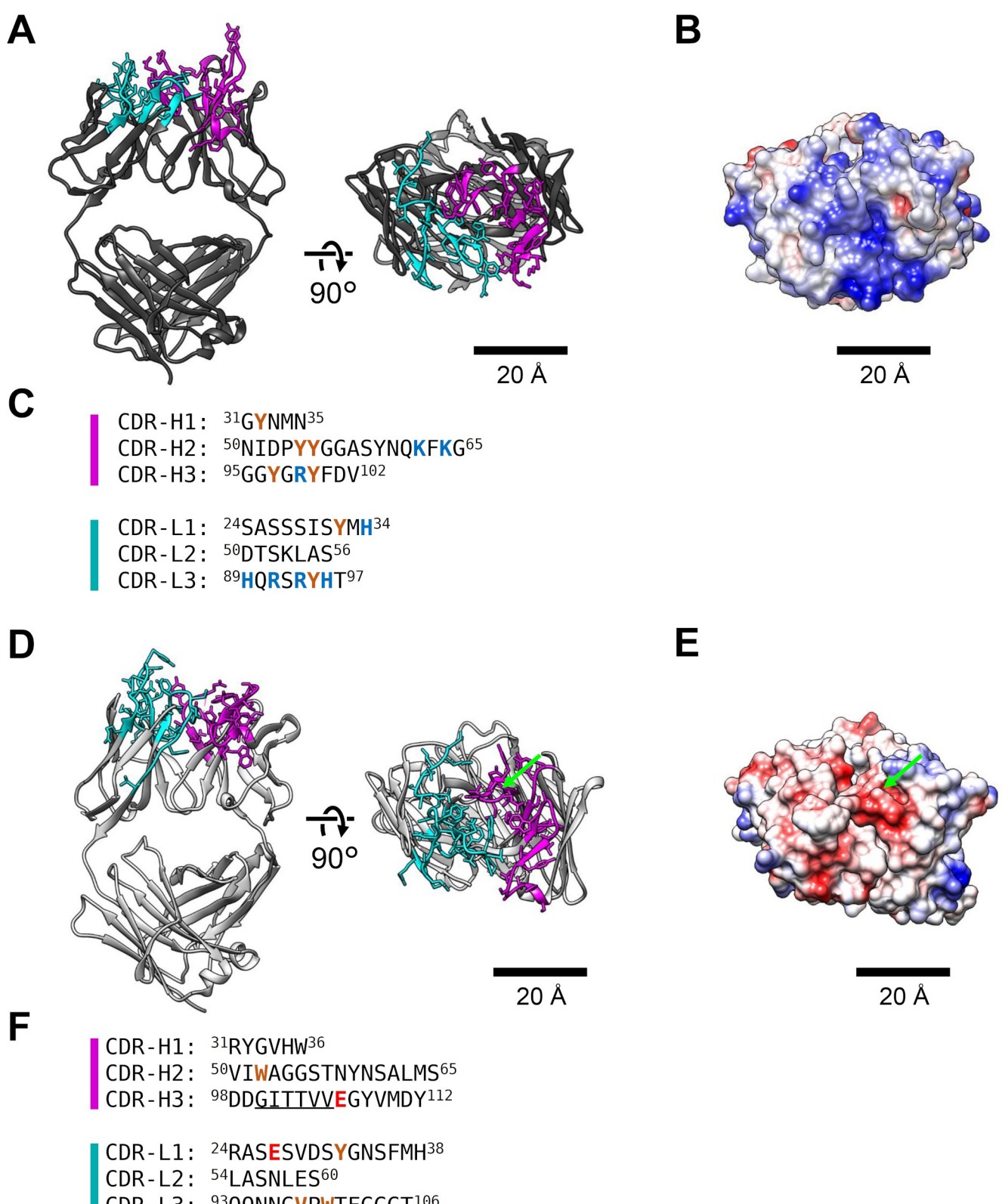

**Fig 3. Crystal structures of the Fab domains of the neutralizing Mabs.** (A) The structure of the anti-gQ1 Fab KH-1, depicted as a *ribbon model*. The CDRs of the heavy chain and light chain are *magenta* and *cyan*, respectively, and the sidechains are shown as *sticks*. (B) The surface model painted by the distribution of the electrostatic potential. The view is the right image of panel A. (C) Sequences of the CDRs. The residues described in the text are in color. (D-F) The same figures for the anti-gH Fab OHV-3. The missing region 100–105 is indicated by green dotted circle in (D) and (E), and by underline in (F).

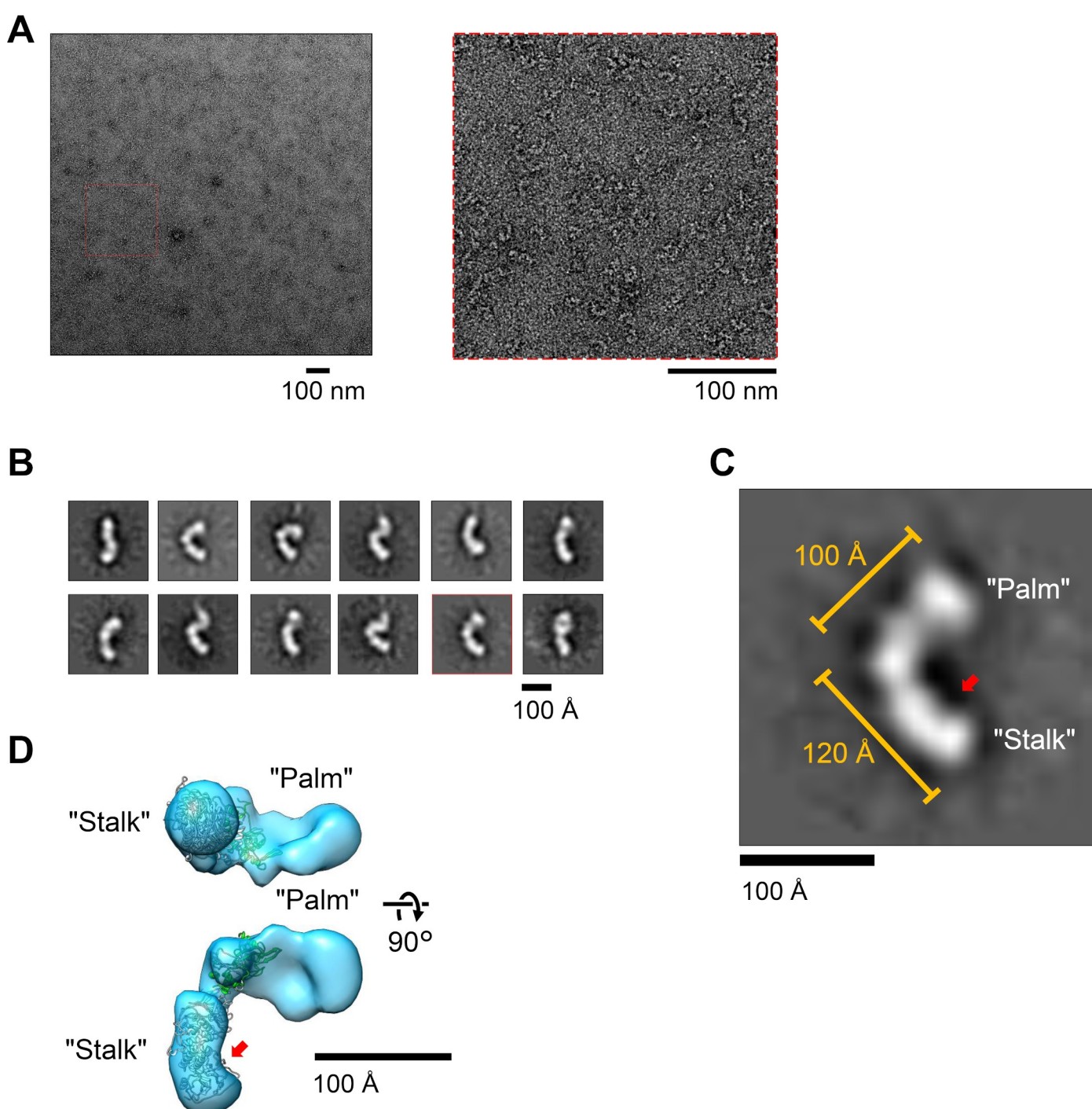

**Fig 4. Negative-staining electron microscopy (EM) analysis of the purified tetramer.** (A) A representative photograph of the purified HHV-6B tetramer by negative-staining EM. The region indicated by a *red rectangle with a dashed line* is enlarged at the *right*. (B) Representative 2D images of the tetramer obtained by single-particle reconstitution by the software Scipion. The *red-boxed* image is shown in more detail in panel (C) A representative image was enlarged to show the structural features. The Stalk and Palm are indicated, and a shallow bend in the stalk is indicated by a *red arrow*. (D) The 3D density of the HHV-6B tetramer. A gH/gL model based on the HCMV gH/gL structure in the pentamer [26] was built by the SWISS-MODEL server [30] and fitted on the density manually with the help of the Fitmap algorithm of the UCSF Chimera program [31]. The parts of the Stalk and Palm and the shallow bend noted in (C) are shown.

With the help of the software program UCSF Chimera [31], we manually fitted a homology model of HHV-6B gH/gL that had been built based on the HCMV gH/gL structure of the pentameric complex [26] on the 3D density (Fig 4D). The size and shape of the gH/gL model were well fitted to the Stalk part of the density, placing the gL model near the base of the Palm part and the C-terminal region of gH at the end of the Stalk side. Approximately one-third of the Palm part was unfilled, and it was accounted for by the gQ1 and gQ2.

### The molecular shape of the immune complex with the Fab domain derived from each neutralizing antibody

Based on the overall shape of the tetramer, we attempted to visualize the binding modes of the two neutralizing Mabs. Two immune complexes of the tetramer and the neutralizing antibodies were analyzed by negative-staining electron microscopy. The Fab domains of the anti-gQ1 Mab KH-1 or the anti-gH Mab OHV-3 were mixed with the tetramer, and then the immune complex was purified by size exclusion column chromatography. The elution volumes shifted slightly to earlier positions for both Fabs, indicating the formation of the immune complexes (Fig 5, S3 Fig).

The single-particle analysis of the tetramer/anti-gQ1 Mab KH-1 Fab complex clearly unveiled extra density at one end of the elongated shape in the 2D averages (Fig 5B), and this was also apparent in the reconstituted 3D density with 51,607 particles at 17.6 Å resolution (Fig 5C), further extending the elongated long axis of the tetramer. The size and shape of the extra density at the tip of the Palm area was consistent with that of the Fab, and the crystal structure shown in Fig 3A could be well-fitted. Because the KH-1 recognizes the gQ1 of the tetramer [27], the overall shape of the immune complex supported the assumed assignment of the Stalk as the gH/gL part and the Palm as the gQ1/gQ2 part.

In contrast, the 2D analysis for the tetramer/anti-gH Mab OHV-3 Fab revealed a relatively compact shape with extra density at a side position against the long axis, showing an additional branch (Fig 5D). The interpretation of the 2D images was difficult compared to that of the tetramer/anti-gQ1 Mab KH-1 complex because the Stalk, Palm and the Fab density could not be distinguished from each other in the 2D images showing multiple knobs. However, the 3D reconstruction with 81,721 particles at 16.8 Å resolution revealed that the Fab density bound to the Stalk part, i.e., the gH/gL area of the tetramer (Fig 5E) which is consistent with the report that OHV-3 recognizes gH [28,29,32].

## Discussion

### The molecular properties of the soluble HHV-6B tetramer

Since the HHV-6B tetramer is the key viral ligand that is essential for the entry of the virus, the present elucidation of the molecular characteristics and interactions with the key molecules including Mabs and the receptor adds to the comprehensive understanding of the entry of viruses and the prevention of this entry, as well as advanced applications for the development of antiviral agents. We have established an expression system for a recombinant, soluble tetramer, and its potential as a vaccine immunogen has been demonstrated [53] in parallel with the present molecular study. The tetramer appeared as a single peak in the size-exclusion column chromatography (Fig 5A and S3A Fig); this was consistent with the monodispersed pattern observed in the EM analysis (Fig 4A). The tetramer was competent in the interaction with the neutralizing Mabs and receptor, and we used the tetramer for the evaluation of the affinities of the neutralizing Mabs and receptor and for the EM analysis in this study. Collectively, our findings highlight the quality of the recombinant tetramer as a useful material for versatile

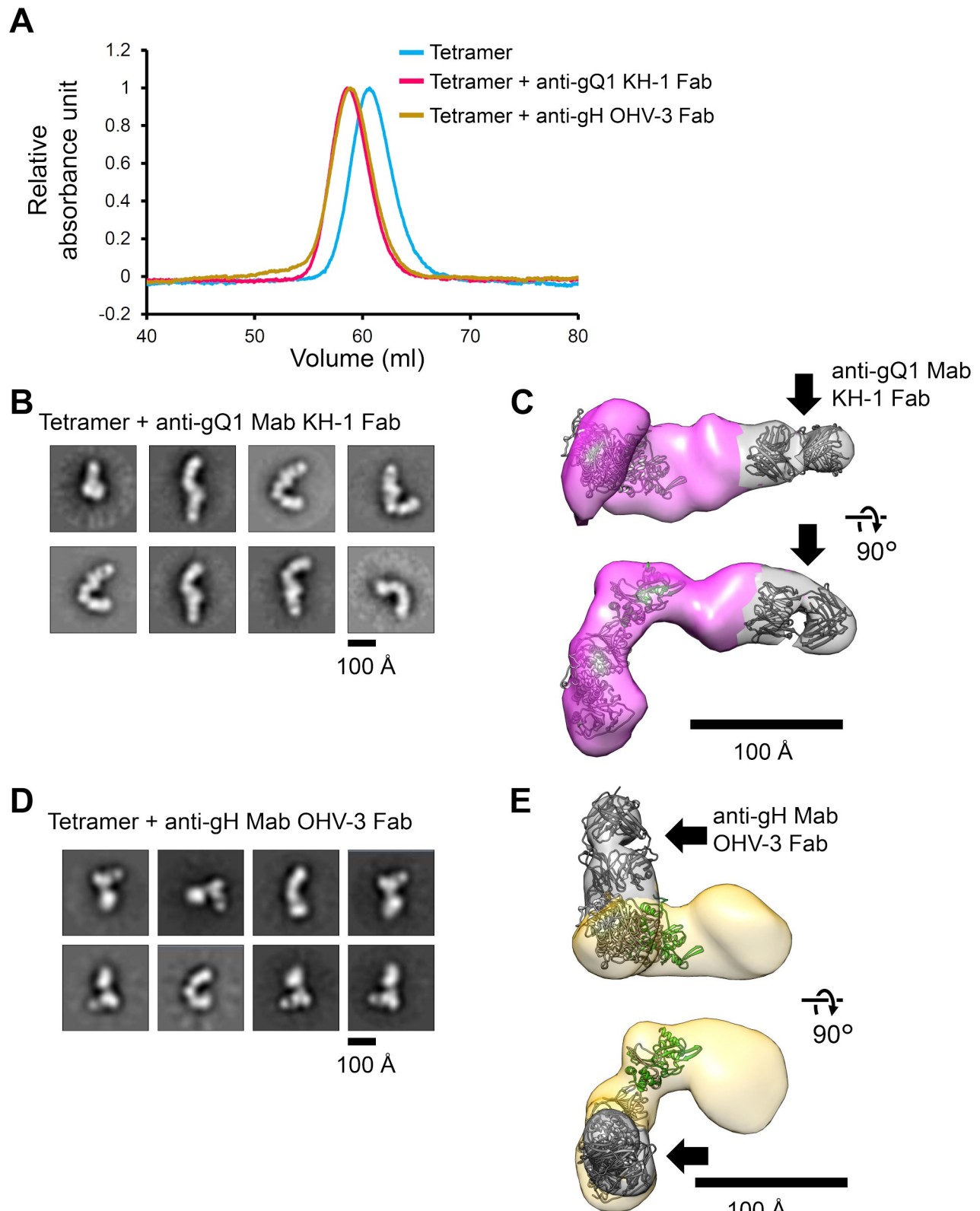

**Fig 5. Negative-staining EM analysis of the purified tetramer with the Fab of each neutralizing Mab.** (A) The elution volumes of the size-exclusion column chromatography experiments are shown for the tetramer (*blue*), the tetramer/anti-gQ1 Mab KH-1 Fab (*red*), and the tetramer/anti-gH Mab OHV-3 Fab (*green*). The full chart is shown in S3A Fig. The absorbance unit values at 280 nm are presented as the relative absorbance

normalized by setting the absorbance value at the peak maximum as 1.0. (B, D) Representative images of the tetramer/anti-gQ1 KH-1 Fab (B) and tetramer/anti-gH OHV-3 Fab (D) obtained by the 2D averaging of classified particles. (C, E) The 3D density reconstructed by the particle reconstitution for the tetramer/anti-gQ1 KH-1 Fab (C) and the tetramer/anti-gH OHV-3 Fab (E). The position of each Fab is indicated by a *black arrow* and the density around each Fab was painted by gray.

research that focuses on viral ligands, such as further structural analyses, evaluations of antibodies and drugs, and applications as vaccine antigens.

## The affinity of the tetramer for the receptor

The interaction between the tetramer and the receptor hCD134 is especially important for both the infection and the host protection against the infection, since the entry of the HHV-6B virion critically depends on the molecular events [18,19]. The results of our SPR analysis revealed the dissociation constant of the interaction between the tetramer and hCD134 as $K_D$ = 18 nM (Table 1, Fig 2C and 2D), which indicates significantly high affinity to realize the firm attachment of the virus on the target cells. This affinity is comparable to those of viral ligands of herpesviruses with their target receptors. The affinity of the EBV gH/gL/gp42 complex with the receptor human leukocyte antigen (HLA) class II molecule DQ2 (HLA-DQ2) was reported to be $K_D$ = 53 nM or 153 nM [33,34]. Fast association and dissociation as observed in the present study (Fig 2C) were also indicated for the interaction between EBV gH/gL/gp42 and HLA-DQ2 [33]. The gH/gL complex of HCMV is associated with gO or UL128/UL130/UL131 and forms a gH/gL/gO trimer or gH/gL/UL128/UL130/UL131 pentamer. The HCMV trimer and pentamer were shown to recognize platelet-derived growth factor-α (PGDFRα) and neuropilin-2 (Nrp2), with affinities of $K_D$ = 2 nM and 300 nM, respectively [35,36]. The considerably high affinity of the HHV-6B tetramer for hCD134 supports the concept of the HHV-6B tetramer's ability to sense activated T cells via hCD134 as the target of infection. It also suggests a requirement of substantial affinity and adequate concentrations of antibodies or antivirals to prevent the ligand-receptor interaction by competition. The further molecular characterization of the tetramer will contribute to a structure-based optimization designed to obtain competing antiviral agents.

## The tetramer's structure

Our EM analysis of the tetramer is the first to unveil the elongated and curved shape of the tetramer (Fig 4). The elongated shape could be interpreted as two parts, i.e., the Stalk and the Palm, which we assigned as gH/gL and gQ1/gQ2, respectively (Fig 4C and 4D). The gH/gL part looked similar to the gH/gL structures of other herpesviruses in size and shape, with a characteristic shallow bend near the expected membrane proximal end of the gH (Fig 4C) [12,22,23,26]. The elongated arrangement of the gQ1/gQ2 in the tetramer is reminiscent of the HCMV pentamer in which the UL128/UL130/UL131 part docks on the gH/gL, extending the longitudinal axis [26,37]. The HCMV pentamer (gH/gL/UL128/UL130/UL131) and the trimer (gH/gL/gO) are located at the associated UL128/UL130/UL131 or gO on the top of gL [37], whereas the EBV gH/gL/gp42 complex does not place the gp42 on gL, but the gp42 almost exclusively interacts with the gH apart from the gL [25,33,34,38]. Thus, the gQ1/gQ2 arrangement unveiled by our present analyses might indicate a shared feature among the viral ligands of betaherpesviruses.

The crystal structure of the HCMV pentamer revealed that the gL of the HCMV pentamer has an N-terminal region forming a base to interact with UL128 and UL131 at the tip of the gH/gL complex, although the N-terminal end folds back to the gH side [26]. The extended N-terminal region is a shared characteristic among the gLs of betaherpesviruses, and it contains

an absolutely conserved cysteine residue that serves to form a disulfide bond with UL128 in the HCMV pentamer and with gO in the trimer [37]. The similar spatial arrangement of gQ1/gQ2 for the gH/gL complex revealed in the present study strongly supports the prediction that the N-terminal region of the HHV-6B gL receives the gQ1/gQ2. Previous research revealed that the HHV-6A gQ2 has disulfide bond(s) with gH/gL [13], and thus gQ2 may form disulfide bond with gL in the tetrameric complex for HHV-6A and HHV-6B.

### The binding modes of the antibodies

The anti-gQ1 Mab KH-1 and anti-gH Mab OHV-3 have neutralizing activity against HHV-6B [27,29], and we recently reported a humanization trial of these murine Mabs as candidate anti-HHV-6B antiviral agents [28]. The present study's SPR analysis revealed the considerable affinities of these Mabs for the soluble tetramer, and the affinities were comparable to those of hCD134. Interestingly, the affinity of the anti-gH Mab OHV-3 ($K_D$ = 2.7 nM) was higher than that of the anti-gQ1 Mab KH-1 (at $K_D$ = 17 nM) even though the neutralizing activity of anti-gQ1 Mab KH-1 against HHV-6B was substantially higher than that of anti-gH Mab OHV-3 [28]. Such an inconsistency between the affinity and neutralizing activity can be explained as the result of the differences in the modes of action. The results of the competition experiment clearly revealed that the anti-gQ1 Mab KH-1 competed with hCD134 in binding to the tetramer, and the anti-gH Mab OHV-3 did not compete even at the concentration range more than the $IC_{50}$ 7.7 μg/ml [28] (Fig 3 and S2B Fig). Because the gQ1 itself is responsible for recognition of the hCD134 [19,20], the simplest explanation is that the binding of anti-gQ1 Mab KH-1 resulted in a sequestration of the hCD134 direct binding site on the tetramer, thereby preventing the viral entry.

Our EM analysis of the tetramer and anti-gQ1 Mab KH-1 complex revealed that the binding site was located on the expected membrane-distal end of the tetramer, further extending the elongated shape (Fig 5D). We thus speculate that the hCD134 binding site on the tetramer is located near the membrane-distal end of the tetramer. It is noteworthy that the tetramer has a critical residue (Glu127) in the gQ1 for the binding to hCD134 [20], and the Lys79 residue in hCD134 is indispensable for the interaction [18], implying a contribution of electrostatic interaction. It is tempting to speculate that the positively charged area observed in the crystal structure of the anti-gQ1 Mab KH-1 (Fig 3B) mimics the electrostatic surface around the Lys79 of hCD134.

Tetramer-hCD134 complex could not be analyzed by negative stain EM in this study because of possible difficulties in sample and experimental technique. It is considered that the relatively small size (approximately 20 kDa calculated from the amino acids sequence) and stick-like shape of the hCD134 molecule [39] might make it difficult to visualize the density in the resolution of negative stain EM analysis. Moreover, the negative staining method might damage the complex as is generally known, considering the relatively unstable nature of the complex indicated by the rapid dissociation parameter in the SPR analysis (Fig 1C and Table 1). The advanced cryoelectron microscopy or the X-ray crystallography are thought to be much more suitable to reveal the tetramer-hCD134 complex, and thus we are going to pursue the issue in future studies.

Both the HCMV pentamer and the HCMV trimer were reported to bind their receptors (i.e., Nrp2 and PDGFRα) at each membrane-distal end, respectively [35,36]. In addition, the Mabs that neutralized HCMV infection to epithelial cells in a pentamer-dependent manner were reported to target the membrane-distal area of the pentamer [37,40]; this is similar to anti-gQ1 Mab KH-1, which targeted the tip of the tetramer (Fig 5D). These findings imply similarity in the receptor binding mode among viral ligands of betaherpesviruses, and they are

in sharp contrast to the case of EBV gH/gL/gp42, which recognizes the receptor near the gH portion due to the gp42 location [25,33,34].

We observed that anti-gH Mab OHV-3 binds at the stalk of the tetramer (Fig 5E) and at a site far from the expected hCD134 binding site, without preventing the binding of hCD134 (Fig 3); this implies that anti-gH Mab OHV-3 may inhibit a tetramer function other than the receptor recognition; for example, the activation of the fusion protein gB at the membrane fusion step [11,12] or the direct inhibition of the fusion function of gH. The binding of anti-gH OHV-3 possibly has an allosteric effect on the hCD134 binding even if their binding sites are distant, however, we could not obtain any data indicating such effect. We also note the difference between the molecular interaction analyzed with purified proteins and actual virus infection in which the interaction occurs on the envelope surfaces with several accompanying factors.

The binding mode between the anti-gH Mab OHV-3 and the tetramer can be further examined based on the combination of the observed EM density, the crystal structure of the OHV-3, and the gH/gL model based on that of the HCMV pentamer (Fig 6). The epitope of the anti-gH Mab OHV-3 has been investigated by Takeda et al. [32]. They revealed that anti-gH Mab OHV-3 recognizes the gH fragment (which consists of residues 272–422), and they identified an arginine residue at 389 as the critical epitope for the interaction. The region 272–422 corresponds to the helix-rich part of domain II (D-II) in the EBV gH domain assignment [24], and our fitting model actually located the region near the Fab density (Fig 6).

Moreover, the critical epitope Arg389 was placed at the base between the gH density and the Fab density. This consistent observation may support the reliability of our model. The membrane-proximal area which consists of the gH is the target of neutralizing antibodies, and a common function of gH has been suggested [12,26,37], although the binding area recognized by anti-gH Mab OHV-3 is unlikely to match that of the well-studied Mab 3G16 (against HCMV gH) [26] or the Mab CL40 (against EBV gH) [34]. The Mab 3G16 binds to the gH at the membrane's proximal end near the C-terminal end of gH, and CL40 binds to the D-II of gH near the N-terminal side from the opposite side (Fig 6B, open arrows). Although any further discussion requires a more concrete model at high resolution, the binding mode of anti-gH Mab OHV-3 recognizing the gH area further supports an additional function of gH.

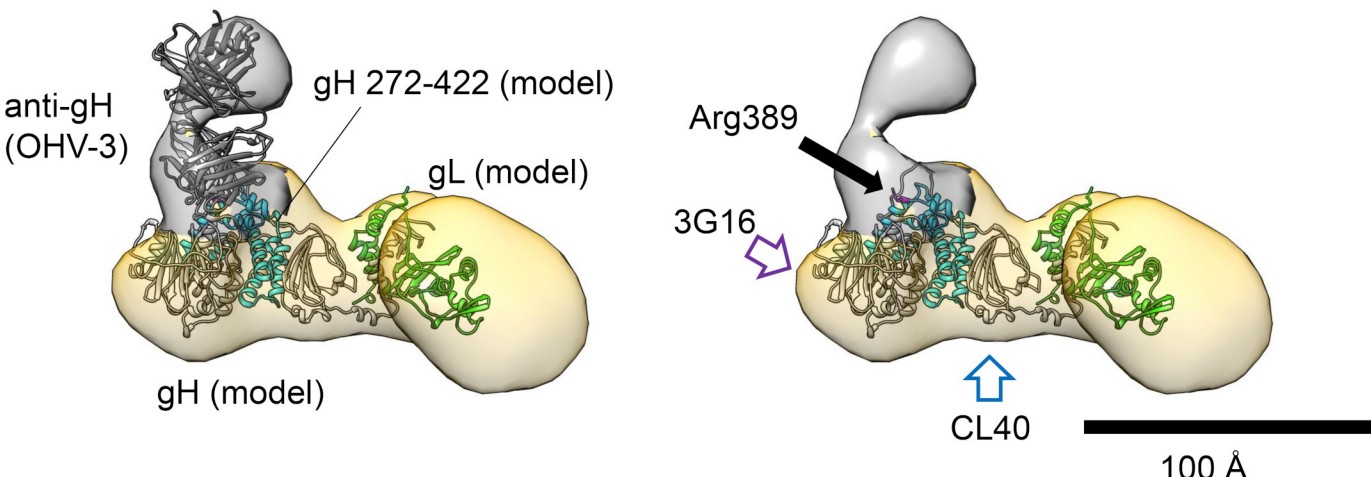

**Fig 6. The epitope of the anti-gH Mab OHV-3.** The gH region 275–422 to which the anti-gH Fab OHV-3 complex can react is colored *cyan* in the gH model. The density around the Fab was colored by gray. To indicate the position of the epitope, the Fab was removed from the right image, and the Arg398 is represented by a *magenta stick*. The approximate locations for the 3G16 binding in HCMV gH and the CL40 binding in EBV gH are indicated by *purple* and *blue open arrows*, respectively.

Collectively, the results of our molecular analysis using the soluble HHV-6B tetramer indicated molecular aspects that are in common with the gH/gL-based viral ligands of other herpesviruses from the macroscopic viewpoint. The characterization of the tetramer as a molecular complex and the formation of immune complexes demonstrated in this study should be followed by further structural analyses using X-ray crystallography and cryoelectron microscopy to unveil the HHV-6B tetramer's unique features and significance in the infection.

## Materials and methods

### Preparation of the tetramer

A mammalian cell based expression system of the HHV-6B tetramer was used in this study. The four genes of tetramer, namely gH, gL, gQ1, and gQ2 derived from HHV-6B HST strain were codon optimized by Invitrogen (Thermofisher Scientific, Waltham, MA, USA). Ectodomain sequence of the gH (residues 16–667) was subcloned into pFuse-hIG1-Fc2 plasmid (InvivoGen San Diego, CA) as described previously [19,41]. Each of the gH ectodomain with human IgG1 Fc and His tag (gHFcHis), gL, gQ1 or gQ2 fragment was subcloned into the pCAGGS-MCS vector [42]. The gL or gQ2 expression cassette including the promoter and polyA sequence from the vector was subcloned into the pCAGGS-gHFcHis or pCAGGS-gQ1 plasmid to construct pCAGGS-gHFcHis/gL or pCAGGS-gQ1/gQ2 plasmid, respectively. Neomycin-resistant gene cassette from pMC1neo-polyA (Agilent Technologies, Waldbronn, Germany) or puromycin-resistant gene cassette from pPur plasmid (Clontech, Mountain View, CA) was inserted into pCAGGS-gHFcHis/gL or pCAGGS-gQ1/gQ2 plasmid, and the resultant vector was named pCAGGS-neo-gHFcHis/gL or pCAGGS-pur-gQ1/gQ2 plasmid, respectively. The sequences of gHFcHis, gL, gQ1 and gQ2 were confirmed by Sanger sequencing with 3130 Genetic Analyzer (Applied Biosystems).

The pCAGGS-pur-gQ1/gQ2 and pCAGGS-neo-gHFcHis/gL plasmids were co-transfected into HEK293S GnTI⁻ cells with Lipofectamin2000 (Invitrogen). The cells were cultured in selection medium with 100 µg/ml of neomycin and 1 µg/ml of puromycin, and viable cells were cloned by endpoint dilution. For the protein expression, the HEK293S GnTI⁻ cells stably harboring the expression vectors pCAGGS-pur-gQ1/gQ2 and pCAGGS-neo-gHFcHis/gL were cultivated in a chemically defined protein-free medium, CD293 medium (Thermofisher Scientific, Waltham, MA, USA) supplemented with 1 µg/ml puromycin and 20 µg/ml gentamicin at 37°C under 5% $CO_2$. The culture supernatant was collected 2 days later, and cell debris was removed by centrifugation. The tetramer with the hFcHis tag in the gH was purified by Ni-NTA agarose resin (Qiagen, Hilden, Germany). HRV-3c protease (Accelagen, San Diego, CA) was used to cleave the hFcHis-tag at 4°C. After the removal of the hFcHis-tag by passing the Protein G Sepharose resin (GE Healthcare Life Sciences, Piscataway, NJ), the sample was subjected to size-exclusion column chromatography equilibrated with a gel filtration buffer (20 mM Tris-HCl, pH 8.2, 100 mM NaCl) and a Superdex 200pg column (GE Healthcare). The purified tetramer was assessed by a sodium dodecyl sulfate-polyacrylamide gel electrophoresis (SDS-PAGE) analysis and a Western blotting analysis as described [27].

### The preparation of the Fab domains of the antibodies

We prepared the anti-gQ1 Mab KH-1 and anti-gH Mab from a hybridoma as described [27,29] and purified them by using Protein A Sepharose resins (GE Healthcare). The Fab domain of each Mab was prepared by papain digestion. The Mabsolution was mixed with a papain agarose resin (Thermofisher Scientific) supplemented with 10 mM cysteine, and the pH was adjusted to approx. pH 7 by adding 1 M Tris-HCl, pH 8.8. After incubation at 37°C

for 48 hr, the papain agarose was filtrated, and the Fc and uncleaved antibody were removed by passing the Protein A Sepharose resins.

## The purification of hCD134-hFcHis

The pCAGGS-MCS [42]-based expression vector of hCD134-hFcHis has been described [19] and used for the preparation of hCD134-hFcHis. Briefly, HEK293T cells were transfected by the vector and cultivated for 2 days in DMEM medium supplemented with 5% fetal bovine serum (FBS) in a $CO_2$ incubator (37˚C, 5% $CO_2$). The hCD134-hFcHis was purified by Ni-NTA agarose.

## Surface plasmon resonance analysis

The surface plasmon resonance experiments were conducted with BiacoreT200 instruments according to the manufacturer's protocol (GE Healthcare). In brief, each of recombinant Protein A (Sigma, St. Louis, MO) or a 1:1 mixture of Protein A and Protein L (Sigma) were immobilized on a CM5 censor chip by an amine coupling method according to the provided protocol (GE Healthcare). The hCD134-Fc was captured via Protein A, and the Mabs were captured via Protein A/L as ligands. The soluble tetramer at a series of concentrations was injected as the analyte, and the response curves were recorded. The signals from a ligand-blank path was subtracted as a baseline. The sensorgrams were analyzed by the software program BIAevaluation (GE Healthcare).

## Competition ELISA

The soluble tetramer in carbonate-bicarbonate buffer was put into each well of a 96-well ELISA-plate (0.3 μg/well) and incubated overnight at 4˚C. The wells were then washed with phosphate-buffered saline (PBS) + 0.02% Tween 20. Each well was blocked by incubation in PBS + 1% bovine serum albumin (BSA) and washed three times with PBS + 0.02% Tween 20.

A series of mixtures of hCD134-hFcHis (0.1 μg/ml) and each Mab at a final concentration of 0.02, 0.038, 0.075, 0.15, 0.3, 0.6, 1.2 or 2.4 μg/ml was prepared and added to each well, incubated for 1 hr, and then washed three times with PBS +0.02% Tween 20. The hCD134-hFcHis in each well was bound by anti-human IgG conjugated with horseradish peroxidase (HRP), and we added a substrate solution for HRP, i.e., 2,2'-Azino-bis (3-ethylbenzothiazoline-6-sulfonic acid) diammonium salt (ABTS) to each well. After a 20 min incubation at room temperature, the HRP reaction was stopped by the addition of 1.5% oxalic acid dehydrate. The absorbance at 405 nm was detected by a multiplate reader Tristar LB941 (Berthold Technologies). The anti-gH Mab bound to the tetramer was detected by using anti-mouse IgG conjugated with HRP instead of the anti-human IgG.

## Crystallization

Each of the Fab solutions at the concentration of 10 mg/ml was used for the crystallization examination by a standard sitting-drop vapor diffusion method. The anti-gQ1 Fab KH-1 was crystallized with the use of a reservoir solution of 100 mM Tris-HCl, pH 8.8 and polyethylene glycol 3350 22.5% (w/v) at 4˚C. The anti-gH Fab OHV-3 was crystallized with the reservoir solution of 100 mM BIS-Tris pH6.5, 0.2 M $MgCl_2$ and polyethylene glycol 3350 22.5% (w/v) at 20˚C.

## Data collection, processing, and structure determination

The X-ray diffraction data were collected on beamline BL26B1 at the SPring-8 synchrotron radiation facility in Harima, Japan [43]. Crystals were soaked in the reservoir solution supplemented

with 25% glycerol and then flash-frozen in liquid nitrogen. The data were processed using the program package XDS [44], and the structure was determined by the molecular replacement method using the software Phenix.phaser [45]. The model was refined by Phenix.refine [46,47] and Coot [48]. Images were prepared by using the UCSF Chimera program [31]. The synchrotron radiation experiments were performed at BL26b1 in SPring-8 with the approval of the RIKEN (Proposal No. 2018B2701 and 2015B2070). The coordinates and structure factors were deposited in the Protein Data Bank under the accession number 6LKT and 6LTG.

### Negative-staining electron microscopy

The purified soluble tetramer and the immune complexes of the tetramer with the Fabs of anti-gQ1 Mab KH-1 or anti-gH Mab OHV-3 were subjected to an electron microscopy analysis. Each solution was loaded on a hydrophilized, carbon-coated copper grid. Uranyl acetate was used for the negative staining. The EM images were recorded by an EM system (JEOL, Tokyo) at ×30,000 magnification and 6.61 Å/pixel resolution in 2,048 × 2,048 dimensions. The EM images were processed by the program suite Scipion ver. 1.1 [49]. The contrast transfer function (CTF) of the images was determined and corrected by GCTF software [50].

The particles were picked by the software RELION [51] and 2D-classified by the the Scipion ver. 1.1 and software cryoSPARC ver. 0.6.5 [52]. The 3D model was reconstituted by the cryoSPARC program, using the Ab-initio Reconstitution method with representative particles (tetramer: 7,709 particles; tetramer/anti-gQ1 Mab KH-1 Fab: 6,460 particles; tetramer/anti-gH Mab OHV-3; 15,902 particles), followed by the homologous refinement method with all particles (tetramer: 65,663 particles; tetramer/anti-gQ1 Mab KH-1 Fab: 51,607 particles; tetramer/anti-gH Mab OHV-3; 81,721 particles). The depiction of the volume data and the model placement were obtained with the UCSF Chimera program [31]. The EM data with the located gH/gL model, anti-gQ1 Mab KH-1 Fab structure, and the anti-gH Mab OHV-3 Fab structure were deposited as a Supporting information S1 data.

## Supporting information

**S1 Fig. Purification of the soluble tetramer.** The tetramer prepared by a mammalian expression system and column chromatography was subjected to the SDS-PAGE analysis and detected by CBB staining (A) and Western blotting (B) with Mabs specific to each component. (TIF)

**S2 Fig. Anti-gH Mab did not compete with the hCD134 binding to tetramer.** (A) The binding of anti-gH Mab was detected by anti-mouse IgG-HRP instead of anti-human IgG-HRP in the same condition as shown in Fig 2B. The open and striped triangles indicate the value without anti-gH Mab and the value without tetramer in presence of anti-gH Mab at 2.4 μg/ml, respectively. The plotted points are the averages of three wells in the same condition. Bars: SD of the wells. One of the duplicated results is shown. (B) The binding of hCD134-hFcHis to the tetramer was detected by the same ELISA experiment as shown in Fig 2B, in a higher concentration range of anti-gH Mab OHV-3. The concentration of the hCD134-hFcHis was 0.1 μg/ml, and the Mab concentration was varied at 0.02, 2.5, 5.0, 20, 40, and 80 μg/ml as the final concentration. The plotted points are the averages of four wells in the same condition. Bars: SD of the wells. The filled arrowheads and open arrowheads indicate the value without Mab and that without hCD134-hFcHis, respectively. The striped arrow head indicated the value for the hCD134-hFcHis binding in the presence of anti-gQ1 Mab KH-1 at 2.4 μg/ml. One of the duplicate results is shown. (TIF)

**S3 Fig. Formation of the immune complexes.** (A) The full chart of the size-exclusion column chromatography experiments are shown. The region from the elution volume 40 ml to 80 ml is shown in Fig 5A. (B) SDS-PAGE analysis of each peak in the panel (A). The band corresponds to the gH, gL, gQ1 and gQ2 are indicated. The bands for Fabs of anti-gQ1 Mab KH-1 and anti-gH Mab OHV-3 are also indicated by *filled arrowhead* and *open allowhead*, respectively.
(TIF)

**S1 Data. The EM data and fitted model.** (A) S1A_Data.mrc: the 3D density obtained by the analysis of tetramer shown in the Fig 4D. (B) S1B_Data.pdb: the HHV-6B gH/gL model built by SWISS-MODEL server [30] fitted to the density of (A). (C) S1C_Data.mrc: the 3D density obtained by the analysis of tetramer + anti-gQ1 Mab KH-1 Fab shown in Fig 5C. (D) S1D_Data.pdb: the HHV-6B gH/gL model and the crystal structure of anti-gQ1 Mab KH-1 Fab fitted to the density of (C). (E) S1E_Data.mrc: the 3D density obtained by the analysis of tetramer + anti-gH Mab OHV-3 Fab shown in Fig 5E. (F) S1F_Data.pdb: the HHV-6B gH/gL model and the crystal structure of anti-gH Mab OHV-3 Fab fitted to the density of (E).
(ZIP)

## Acknowledgments

We thank Dr. Hiroki Tsuruta (Center for Applied Structural Science, Center for Collaborative Research and Technology Development, Kobe University) for the technical support in X-ray crystallography.

## Author Contributions

**Conceptualization:** Mitsuhiro Nishimura, Yasuko Mori.

**Data curation:** Yasuko Mori.

**Formal analysis:** Mitsuhiro Nishimura.

**Funding acquisition:** Mitsuhiro Nishimura, Yasuko Mori.

**Investigation:** Mitsuhiro Nishimura, Bernadette Dian Novita, Takayuki Kato, Lidya Handayani Tjan, Bochao Wang, Aika Wakata, Anna Lystia Poetranto, Akiko Kawabata, Huamin Tang, Taiki Aoshi.

**Methodology:** Mitsuhiro Nishimura, Takayuki Kato, Akiko Kawabata, Huamin Tang, Yasuko Mori.

**Project administration:** Yasuko Mori.

**Software:** Mitsuhiro Nishimura, Takayuki Kato.

**Supervision:** Yasuko Mori.

**Visualization:** Mitsuhiro Nishimura.

**Writing – original draft:** Mitsuhiro Nishimura.

**Writing – review & editing:** Yasuko Mori.

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
