## [Decision Letter · Decision Letter 0]

5 Mar 2020

Dear Prof. Mori,

Thank you very much for submitting your manuscript "Structural basis for the interaction of human herpesvirus 6B tetrameric glycoprotein complex with the cellular receptor, human CD134" for consideration at PLOS Pathogens. As with all papers reviewed by the journal, your manuscript was reviewed by members of the editorial board and by several independent reviewers. In light of the reviews (below this email), we would like to invite the resubmission of a significantly-revised version that takes into account the reviewers' comments.

We cannot make any decision about publication until we have seen the revised manuscript and your response to the reviewers' comments. Your revised manuscript is also likely to be sent to reviewers for further evaluation.

Sincerely,

Ekaterina E. Heldwein

Guest Editor

PLOS Pathogens

Klaus Früh

Section Editor

PLOS Pathogens

Kasturi Haldar

Editor-in-Chief

PLOS Pathogens

orcid.org/0000-0001-5065-158X

Michael Malim

Editor-in-Chief

PLOS Pathogens

orcid.org/0000-0002-7699-2064

Reviewer's Responses to Questions

**Part I - Summary**

Reviewer #1: This work provides the first structural depiction of the HHV6 gHgLgQ1gQ2 tetramer, a complex that is required for HHV6 entry into cells. The authors characterize the tetramer binding to receptor and to neutralizing antibodies (nMAbs) and report the EM structure of the tetramer, with or without nMAbs bound.

The authors purified soluble tetramer expressed in mammalian cells (Fig S1). They used SPR to examine tetramer binding to two nMAbs and receptor (Fig 1, Tbl 1). Tetramer affinity for all three was high and, interestingly, the receptor binding had a fast association and dissociation. Using competition ELISA, the anti-gQ1 nMAb was shown to block tetramer-receptor binding, whereas anti-gH nMAb did not (Fig 2), indicating that the two nMAbs employ distinct mechanisms of neutralization. The crystal structures of unliganded Fabs for both nMAbs were solved (Fig 3, Tbl 2). Single particle EM analysis of the tetramer revealed an elongated curved structure with “palm” and “stalk” sites (Fig 4). Manually fitting an gHgL model (based on HCMV gHgL) into the density placed gH in the stalk and gL near the palm. Most of the palm was left unfilled, suggesting that gQ1 and gQ2 reside in the palm. Complexes of tetramer plus each Fab were purified (Fig S2). Single particle EM analysis of these Fab-tetramer complexes showed two distinct sites of Fab binding and supported the earlier manual fitting of the gHgL structure in the density (Fig 5). The anti-gQ1 Fab binding was at the end of the structure, whereas the anti-gH Fab bound to the side the structure.

The work is convincing and logical. The paper is well written and concise. This work will have significant impact in the field of herpesvirus entry into cells. The biochemical (ELISA and SPR) and structural (crystallography and EM) approaches provide strong evidence that receptor binding occurs at gQ1, at the distal end of the tetramer. By following up their fitting of a gHgL model into their tetramer EM density with binding studies of the two Fabs, the work is persuasive that this fitting is accurate. gQ1 and gQ2 appear to dock at the gL tip of gHgL, similar to the location of gO in HCMV trimer and UL128/UL130/UL131 in HCMV pentamer. Thus, this work demonstrates that gHgL complexes from both betaherpesviruses are similar to one another, but quite different than the EBV gHgLgp42 complex, in which gp42 binds along the side of gH. Identifying the structural location of the anti-gH Fab binding is of particular interest because the mode of neutralization for this Fab is unknown (and not due to interference with receptor binding).

Reviewer #2: Nishimura et al. report the modelled structure of the HHV-6B gH/gL/gQ1/gQ2 (tetramer) complex alone and complexed with the Fab fragments of two monoclonal antibodies to gQ1 and gQ2. The authors also determined the affinity of the two mAbs as well as of the CD134 receptor for the tetramer and demonstrate direct inhibition of tetramer binding by the gQ1 antibody KH-1 but not by the gH antibody OHV-3. Using negative-staining EM, averaging and 3D modelling, the authors provide an informative structure of the tetramer complex more or less at the domain level and of the tetramer complexed with each of the two Fabs, fitting the crystal structure of the two Fabs, which they also determined, into the EM model.

The manuscript is generally well written.

The crystal structures of the Fab domains of the variable antibodies provide potentially useful information for further targeted development and optimization of these antibodies.

The overall shape of the tetramer complex is highly interesting to the field.

One question is obvious: Why was the same EM analysis not attempted in complex with CD134, in addition to the Fab fragments? This would have been even more informative.

**Part II – Major Issues: Key Experiments Required for Acceptance**

Reviewer #1: (No Response)

Reviewer #2: As the structural models for the tetramer built in this paper are not illustrative but constitute basically the main piece of novel information contributed by this study, apart from the measured affinities, these models should be deposited in a public database or at least be made available as supplemental information so that the community can use them for visualization or for the informed design of experiments.

Line 154 (Wang et al. manuscript submitted) I am not sure what the regulations on these types of citations are. If it is not a preprint, can it be cited? Also what type of manuscript is this, is there some sort of redundancy between the manuscripts? I personally don’t like citations to something that can’t be read - so if it is cited, it should be made available to the editor and the reviewers. Publication as a preprint would be an option, too.

Fig 2. It would be highly interesting for the reader to see how the inhibitory concentrations in the ELISA assay compare to the inhibitory concentrations in infection experiments for the anti-gQ1 antibdoy KH-1. Aaccording to the 2019 Wang paper, the IC50 for the gQ mAb is 0.17µg/ml, for the gH mAb ~40fold higher at 7.7 µg/ml. The inhibition curve here for KH-1 indicates that IC50 for inhibition of binding of CD134 is higher than for infection. While the concentration range around the IC50 in infection assays is tested for the gQ1 mAb KH-1, it is not tested for the gH mAb OHV-3 in this competition assay. It could therefore be that OHV-3 also inhibits CD134 binding via an allosteric effect on the tetramer. Even if unlikely, this can’t be ruled out because the concentration range where an inhibitory effect on infection was observed is not reached in the experiment. The experiment should be repeated using higher concentrations of OHV-3, or the possibility of allosteric inhibition should at least be discussed. It may seem unlikely, as the affinity as measured by biacore of OHV-3 was higher than that of KH-1. But affinity by biacore, affinity on the virus surface, and affinities on ELISA plates, as well as IC50 in infections may not be directly comparable. These possibilites should be discussed in lines 417-431, line 451. Alternatively, experimental evidence of an OHV-3-tetramer-CD134 complex e.g. by immunoprecipitation of tetramer-CD134 complex would support the authors’ interpretation and rule out allosteric inhibition of receptor binding, and would clarify the mode of action of OHV-3. This is actually an easy experiment and would strongly support the author’s conclusions. Or maybe this has already been done (not to my knowledge) and could be cited?

Line 288: please explain “some of the 2D averaged…”. What percentage? Could this be an entirely different conformation? If this “spiral shape” is mentioned, it should be illustrated properly – which of the micrographs displays this shape?

**Part III – Minor Issues: Editorial and Data Presentation Modifications**

Reviewer #1: - The authors describe a spiral structure (line 228) observed in the 2D EM images of the tetramer, but they do not show a picture of that structure. They could consider adding a sample spiral image to Fig. 4 (or as a supplemental figure), in case this structure is of interest to others in future work.

- The authors describe that interpretation of the 2D images for the anti-gH Fab was more difficult than for the anti-gQ1 Fab (line 342). They could expand on this in a sentence to explain why.

- In Fig 1, the authors could label figure panels with the analyte and immobilized molecule. The information is in the figure legend, but labeling the graphs would be helpful for quick comprehension. Part D x axis could read “concentration of tetramer (nM)”.

- In Fig 2, the authors could add names of Abs to graphs so reader doesn’t have to refer to the legend to understand the result immediately.

- In Fig 3, the green dotted line isn’t clear to see, but this isn't critical.

- In Fig 5, labeling the figure with the names of the Fabs would save the reader time. Coloring gHgL in the density would help distinguish it from the Fab, especially in part E.

Reviewer #2: Line 39: please explain “significantly high” or rephrase.

Fig 2.: The concentrations of the antibodies should also be given in nM, for better comparison to figure 1.

Line 398-402: I am not sure that I read this sentence correctly, and I am not sure that I understand what exactly it means. It should be rephrased. Also, I think that a figure would be much more helpful to illustrate the similarities and differences between the different gH/gL/… complexes. Again, the models should be made available for viewing in a molecule viewer, which would make comparisons easier.

Line 411-414: I understand the Akkapeiboon reference in a way that there is experimental evidence for a gL-gQ2 disulfide bond for HHV-6A, so that is not exactly a prediction, but rather fits with the data for HHV-6A.

PLOS authors have the option to publish the peer review history of their article (what does this mean?). If published, this will include your full peer review and any attached files.

Reviewer #1: Yes: Sarah Connolly

Reviewer #2: No
---

## [Editor Report · Decision Letter 1]

20 May 2020

Dear Prof. Mori,

We are pleased to inform you that your manuscript 'Structural basis for the interaction of human herpesvirus 6B tetrameric glycoprotein complex with the cellular receptor, human CD134' has been provisionally accepted for publication in PLOS Pathogens.

Best regards,

Ekaterina E. Heldwein

Guest Editor

PLOS Pathogens

Klaus Früh

Section Editor

PLOS Pathogens

Kasturi Haldar

Editor-in-Chief

PLOS Pathogens

orcid.org/0000-0001-5065-158X

Michael Malim

Editor-in-Chief

PLOS Pathogens

orcid.org/0000-0002-7699-2064
---

## [Editor Report · Acceptance letter]

30 Jun 2020

Dear Prof. Mori,

We are delighted to inform you that your manuscript, "Structural basis for the interaction of human herpesvirus 6B tetrameric glycoprotein complex with the cellular receptor, human CD134," has been formally accepted for publication in PLOS Pathogens.

Best regards,

Kasturi Haldar

Editor-in-Chief

PLOS Pathogens

orcid.org/0000-0001-5065-158X

Michael Malim

Editor-in-Chief

PLOS Pathogens

orcid.org/0000-0002-7699-2064